# Bispecific Antibodies in Lung Cancer: A State-of-the-Art Review

**DOI:** 10.3390/ph16101461

**Published:** 2023-10-14

**Authors:** Atulya Aman Khosla, Karan Jatwani, Rohit Singh, Aswanth Reddy, Ishmael Jaiyesimi, Aakash Desai

**Affiliations:** 1Division of Internal Medicine, William Beaumont University Hospital, Royal Oak, MI 48073, USA; atulya.khosla@corewellhealth.org; 2Division of Hematology-Oncology, Roswell Park Cancer Center, Buffalo, NY 14203, USA; 3Division of Hematology-Oncology, University of Vermont, Burlington, VT 05405, USA; 4Division of Hematology-Oncology, Mercy Clinic, Fort Smith, AR 72903, USA; 5Division of Hematology-Oncology, William Beaumont University Hospital, Royal Oak, MI 48073, USA; 6Division of Hematology and Oncology, Department of Medicine, University of Alabama at Birmingham, Birmingham, AL 35233, USA

**Keywords:** bispecific, antibodies, lung cancer, NSCLC, novel therapies, immunotherapy, solid tumors

## Abstract

Bispecific antibodies have emerged as a promising class of therapeutics in the field of oncology, offering an innovative approach to target cancer cells while sparing healthy tissues. These antibodies are designed to bind two different antigens, enabling them to bridge immune cells with cancer cells, resulting in enhanced tumor cell killing and improved treatment responses. This review article summarizes the current landscape of bispecific antibodies in lung cancer, including their mechanisms of action, clinical development, and potential applications in other solid tumor malignancies. Additionally, the challenges and opportunities associated with their use in the clinic are discussed, along with future directions for research and development in this exciting area of cancer immunotherapy.

## 1. Introduction

Lung cancer is the leading cause of cancer-related mortality in the United States, accounting for 23% of all cancer deaths [1]. It is classified broadly by morphologic subtypes: non-small cell lung cancer (NSCLC), which includes adenocarcinoma, squamous cell carcinoma, large cell cancer, and adenosquamous carcinoma; and the less common subtype, small cell lung cancer (SCLC) [2]. There has been tremendous progress in developing targeted therapies and immunotherapies to treat advanced-stage NSCLC [3]. In the absence of a targetable driver mutation, metastatic and advanced NSCLC and SCLC are treated with a combination of chemotherapy and immunotherapy [4,5].

Monoclonal antibodies have been utilized to treat NSCLC and have led to improved response and survival [6]. With the success of anti-PD-1/PD-L1 monoclonal antibody immunotherapies and, more recently, antibody drug-conjugates [7], there is an increasing interest in utilizing antibody-based therapies for the treatment of lung cancer. This has resulted in the development of bispecific antibodies (bsAbs), which have two distinct antigen binding sites capable of binding two different antigens or epitopes on the same antigen.

Although the concept of bispecific antibodies was first proposed by Alfred Nisonoff and colleagues back in 1961, their clinical relevance has recently come to light after the FDA approval of blinatumomab for the treatment of acute lymphoblastic leukemia [8,9]. Depending upon the type of cell and genetic engineering platform utilized for their synthesis, bispecific antibodies can exhibit different formats and mechanisms of action. This review article discusses different bispecific antibody structures and mechanisms of action and summarizes their clinical application and ongoing evaluation in solid tumors with a particular focus on thoracic malignancies.

## 2. Structure of Bispecific Antibodies

While natural antibodies have two targeting arms that bind to the same target antigen, bispecific antibodies are engineered hybrid molecules with two distinct binding domains that target two distinct antigens. Bispecific antibodies are divided into two categories based on the Fc region: IgG-like subtypes containing an Fc region and non-IgG-like subtypes without an Fc region. (Figure 1) [10].

### 2.1. Fc-Bearing Subtypes

The IgG subtype class of bispecific antibodies possesses an Fc region and has the advantages of better stability and solubility, a longer half-life, and the ability to kill cancer cells via complement-dependent and antibody-dependent cytotoxicity. Several methods are utilized in bsAbs production, and methods such as Knobs-into-holes, CrossMab, and DuoBody utilize a symmetrical structure that enables stability similar to natural IgG antibodies. However, the proximity of antigen-binding sites may lead to reduced functional potency. In contrast, the asymmetric nature of other methods resolves this issue and allows for the monovalent binding of CD3, thus reducing the toxicity of CD3 antibodies when binding various tumor antigens [11]. Despite advances in formulation techniques, the difficulty in its design and preparation remains a limitation.

#### 2.1.1. IgG-like

##### Knobs-into-Holes

The Knobs-into-holes method was adopted by Genentech and led to the production of >90% of desired product, enabling large-scale manufacturing [12]. It is made by engineering the CH3 domain of the antibody by replacing certain amino acids to create a “knob” or “hole” on the two heavy chains to promote heterozygous dimerization [13]. This technology prevents the issue of heavy-chain mismatches, but inaccurate light-chain binding leads to light-chain mismatches [14].

##### CrossMab

The CrossMab variant is based on the inter-arm cross-exchange of bispecific IgG antibodies and heterodimerization of heavy chains via knob-into-hole configuration, thereby ensuring correct chain binding [15]. This production technology, developed by Roche, has proven effective so far, with over eight bispecific antibodies developed by this technology currently in clinical development. Fragments may be exchanged at the level of Fab domains, variable VH-VL domains, and invariant CH1-CL domains [16].

##### DuoBody

This technology of producing bispecific antibodies was developed by Genmab and involved a controlled Fab-arm exchange redox reaction of two parental homo-dimeric IgG1 monoclonal antibodies. The parental antibodies undergo a selective reduction of disulfide bonds at the hinge region, and the chains are recombined through amino acid substitutions to form a bispecific antibody, thereby maintaining the structural integrity and Fc fragment function of the homo-dimeric monoclonal antibody [17]. The bispecific antibody amivantamab developed using this platform targets epidermal growth factor receptor (EGFR) and mesenchymal-epithelial transition (MET) to treat NSCLC.

##### Triomab Quadroma

TriOn Pharma invented this technology to produce bispecific antibodies with the fusion of two different hybridoma cells to generate monoclonal antibodies. Due to the random assembly of heavy and light chains, a quadroma produces multiple antibody structures (including non-functional), but only one is the obligatory bifunctional antibody [18]. Catumaxomab is a bispecific antibody made using this technology, which targets EpCAM and CD3 and consists of mouse IgG2a and rat IgG2b [19].

#### 2.1.2. IgG-Modified

##### Dual-Variable Domains Ig (DVD-Ig)

The DVD-Ig technology was developed by Abbott and aims to prevent any mismatches between the heavy and light chains. It consists of a tetravalent IgG-like molecule containing an Fc region, and each arm of the antibody comprises two variable domains (VDs), consisting of an external VD (VD1) composed of VH1 and VL1 and an internal VD (VD2) composed of VH2 and VL2 [20].

### 2.2. Fc-Free Fragment-Based Subtypes

The non-IgG fragment-based bispecific antibodies are based on the structure of a single-chain variable fragment (scFv), a genetically engineered antibody with VL and VH regions, and a peptide linker of amino acids [21]. Due to the lack of an Fc fragment, this subtype achieves superior tissue penetration, shows weak immunogenicity, and enables the administration of a low drug dose. The limitations inherent to its small structure include an unstable structure with no Fc fragment, a short half-life, and low expression [22].

#### 2.2.1. Bispecific T-Cell Engager (BiTE)

BiTE is a novel subtype of bispecific antibodies and consists of two binding domains that link endogenous T-cells to tumor cells. The binding domains consist of two single-chain variable fragment chains from a monoclonal antibody that bind to the tumor antigen and CD3 on T-cells, respectively, and are linked by a flexible peptide chain [23]. Upon binding to both of these sites, T-cell proliferation ensues, increasing the number of effector cells and leading to the effective lysis of cancer cells [24]. BiTE constructs have the unique ability to engage any type of T-cell since the interaction does not involve co-stimulation or necessitate the involvement of a major histocompatibility complex [25]. Blinatumomab was the first BiTE molecule to be approved for the treatment of relapsed or refractory B-cell precursor acute lymphoblastic leukemia. It targets CD19 on tumor cells and links it to CD3 on the T-cell receptor, thereby exerting its effect independent of genetic alterations [26].

#### 2.2.2. Dual-Affinity Re-Targeting Molecules (DART)

DART is a type of bispecific antibody construct developed by MacroGenics. It comprises a linker that connects an antibody’s VL and VH sequence with another to form an scFv and express two unique antigen-binding sites. The DART construct mimics the natural interaction within the IgG molecules, which makes it different from BiTE. The presence of a disulfide bond formed due to the presence of cysteine at the C-terminus further enhances the stability of the overall construct [27]. Flotetuzumab is a DART molecule made available in Europe and Japan as a rescue immunotherapy for patients with refractory acute myeloid leukemia. It targets CD3 on T-cells and CD123 on the myeloid tumor cells [28].

#### 2.2.3. Tandem Diabodies (TandAb)

TandAb is a technology platform that forms tetravalent bispecific antibodies capable of binding two sites for each antigen. It was developed by Affimed, and the chain is constructed in such a manner that the N-terminus to C-terminus is arranged as VL1-VH2-VL2-VH1, with the two peptide chains pairing in opposite directions [29]. Tandem Diabodies can recruit both T and NK cells and utilize both adaptive and innate immune systems. AFM13 is a type of TandAb construct that targets CD30, present on lymphoma cells, and CD16a on NK cells, and has shown efficacy in managing Hodgkin lymphoma [30]. In a single-center phase I/II trial, AFM13 was combined with cord blood-derived NK cells in patients with relapsed/refractory CD30+ lymphoma. In 30 treated patients and a median follow-up of 8 months, the event-free survival (EFS) and OS were 57% and 83%, respectively [31].

#### 2.2.4. Bispecific Nanobody (BsNb)

The BsNb structure refers to a construct developed by Ablynx that contains only heavy chains and preserves the VH region through recombinant technology. The aim is to connect the VH regions of two or more antibodies to lead to multi-specific binding while retaining the advantages of having a small molecular weight and considerable tissue penetration [32]. A preclinical study with a novel BsNb targeting PDL1 and CXCR showed antitumor activity against pancreatic cancer cells, possibly by the requirement of cytotoxic T cells [33].

## 3. Application in Lung Cancer

The success of bispecific antibodies in hematologic malignancies has led to their development and application in solid tumors. To date, the FDA has approved eight bispecific antibodies in oncology, with over 200 in various stages of development (Table 1) [34,35,36]. Bispecific antibodies are engineered to bind to two different antigens or two different epitopes on the same antigen. Their mechanisms of action vary according to the epitopes/antigens recognized. Figure 2 is a schematic representation of the target areas of various bispecific antibodies. However, four major mechanisms are exploited by the majority of bispecific antibody molecules used in clinics today. This includes the recruitment and activation of immune cells to exert anti-tumor activity (such as bispecific T cell engagers), the blocking of dual signaling pathways, the blocking of immune checkpoints, or the forced association of protein complexes. There are several bispecific antibodies currently in trials evaluating their safety and efficacy in lung cancer (Table 2).

### 3.1. MET Targeted

#### 3.1.1. Amivantamab

Amivantamab is the only bispecific antibody approved by the FDA for patients with NSCLC specifically harboring an EGFR exon 20 insertion mutation [37]. EGFR mutations occur in 15–20% of all NSCLCs and lead to constitutively activated signaling pathways, and MET exon 14 alterations are found in 3–4% of NSCLCs [38,39]. c-MET amplification mutation creates a pathway that bypasses EGFR, thereby resulting in resistance to EGFR-tyrosine kinase inhibitors (TKIs) [40]. Amivantamab targets both EGFR and MET genes by binding to the extracellular domains and blocking their respective ligand–receptor interaction. It also induces the degradation of the receptors in vivo [41]. Finally, it can bind to immune effector cells via ADCC, thereby eliminating antigen-expressing tumor cells (Figure 2) [42].

The CHRYSALIS Phase I study utilized amivantamab in EGFR exon 20 insertion-mutated NSCLC progressing on platinum chemotherapy and reported an overall response rate (ORR) of 40% and a median duration of response (DoR) of 11.1 months [43]. The updated results of the CHRYSALIS study found that following progression on platinum-based treatment, 114 patients with non-small cell lung cancer (NSCLC) with EGFR Exon 20 insertion mutations had a median OS of 23 months (95% CI 18.5–29.5). The overall response rate (ORR) was 37% (95% CI 28–46), the median duration of response was 12.5 months (95% CI 6.9–19.3), and the median progression-free survival was 6.9 months (95% CI 5.6–8.8) after a median follow-up of 19.2 months. Rash (89%) and infusion-related events (67%) were the most common all-grade toxicities recorded.

The CHRYSALIS-2 study, which was reported at ASCO 2023, investigated the effectiveness of amivantamab and lazertinib (3rd generation EGFR TKI) in patients with EGFR-mutated NSCLC who had disease progression on Osimertinib or chemotherapy-naive patients with EGFR exon 19 deletion or L858R mutation. It had a 36% ORR with a median response time of 9.6 months. In a subgroup examination of individuals with MET mutations, the ORR was 61% for MET + ve mutations and 12% for MET-ve mutations. The median PFS for MET + ve was not attained, whereas PFS for MET-ve was 4.1 months. According to the findings, the MET + ve mutation may be a predictive biomarker for response to amivantamab plus lazertinib (3rd generation EGFR TKI) in the post-osimertinib, chemotherapy-naive group [44]. The MARIPOSA trial is a phase III study currently underway that is evaluating the efficacy of amivantamab with lazertinib compared to osimertinib, a well-established third generation TKI in the treatment of EGFR-mutant (exon 19 deletions or exon 21 L858R substitution) NSCLC (NCT04487080), (Table 2) [45].

#### 3.1.2. Bafisontamab (EMB-01)

EMB-01 is an innovative EGFR and c-MET targeting bispecific antibody with a mechanism of action similar to amivantamab. It was developed by Epimab using a Fabs-In-Tandem Immunoglobulin (FIT-Ig) platform. A phase I/II trial evaluating its efficacy in advanced solid tumors is currently underway (NCT03797391) [46]. Preliminary phase I results of this trial reported a manageable safety profile, a disease control rate of 42.1%, and a 1600 mg IV weekly as the recommended phase II dose [47]. The phase II portion of the trial enrolls advanced/metastatic NSCLC with EGFR mutation (including T790M) and/or MET amplification progressed or intolerant to standard therapy (NCT05498389).

### 3.2. DLL3-Targeted

#### Tarlatamab (AMG 757)

Tarlatamab is a BiTE bispecific antibody that targets delta-like ligand 3 (DLL3), which is expressed on the surface of tumor cells and in over 80% of SCLC, with a low to null expression in normal lung tissue. It is currently being studied in a Phase II study for treating relapsed/refractory SCLC after two or more lines of treatment (NCT05060016) and a Phase III study of tarlatamab compared with standard of care in patients with relapsed SCLC after platinum-based first-line chemotherapy [48]. This is based upon encouraging results from a phase I study of AMG 757, which demonstrated confirmed responses in 23% of patients and a median DOR of >12 months, median OS of 13.2 months, and an acceptable safety profile with doses up to 100 mg [49]. An ongoing trial is also looking into combining tarlatamab with immunotherapy in SCLC (NCT03319940).

### 3.3. EpCAM Targeted

#### 3.3.1. Catumaxomab

Catumaxomab is a triomab that targets the EpCAM protein and CD3 on T cells and was approved in the past by the European Union for malignant ascites due to epithelial tumors [19]. EpCAM is a type I transmembrane protein primarily considered an adhesion molecule; however, studies have shown diverse biological functions, including regulation of cell proliferation and a biomarker of cancer stem cells and circulating tumor cells. It has been recognized as a highly expressed marker in carcinomas associated with a poor prognosis [50]. Went et al. demonstrated a considerable expression of EpCAM in lung cancers, reaching up to 65%, and a negative correlation with survival time [51]. A phase I trial of catumaxomab in patients with NSCLC confirmed 5 μg as a maximum tolerated dose with a pre-medication of 40 mg dexamethasone [52]. The drug is no longer available for use as it was withdrawn from the market due to commercial issues.

#### 3.3.2. Solitomab (MT110, AMG 110)

Solitomab is an EpCAM/CD3 BiTE bispecific antibody demonstrated to stimulate CD4 and CD8 positive T-cells and reactivate tumor-resident T-cells to deplete cancerous cells [53,54]. Across a phase I dose-escalation study, solitomab was administered to a cohort of 65 patients with recurrent solid cancers, including lung cancer (*n* = 9), and an MTD of 24 μg was found. Due to the shorter half-life, solitomab was administered as a continuous infusion over 4 weeks. 95% of patients had grade ≥ 3 adverse effects (AEs), predominantly diarrhea, elevated transaminases, and increased lipase [55].

### 3.4. HER2/HER3 Targeted

#### 3.4.1. Zenocutuzumab (MCLA-128)

Zenocutuzumab is an IgG-based bispecific antibody that targets NRG1 fusion and blocks NRG1fusion-HER3 interaction and HER3/HER2 heterodimerization. NRG1 fusions are reported in about 20% of non-mucinous and up to 30% of mucinous lung cancers. The activation of HER receptors leads to downstream signaling through the MAPK/ERK and PI3K-AKT pathways, causing tumor cell proliferation. Preclinical studies have evaluated the effectiveness, pharmacokinetics, and pharmacodynamics of zenocutuzumab to determine the starting dose for the clinical studies [56,57]. The eNRGy trial is a phase I/II dose escalation trial to determine the efficacy of zenocutuzumab in solid tumors harboring an NRG1 fusion (NCT02912949). Part 2 of this trial included a separate cohort of NSCLC patients with NRG1 fusion. As of January 2022, results were evaluated in 71 patients with measurable disease. ORR was reported as 34% in all patients and 35% in NSCLC patients [58].

#### 3.4.2. MM-111

MM-111 is a novel bispecific antibody made with modified human serum albumin that targets the HER2/HER3 heterodimer and inhibits downstream signaling. In the phase I trial, MM-111 monotherapy was utilized in patients with HER2-amplified solid tumors, and MTD was not reported, as no patients experienced dose-limiting toxicity (DLT) (NCT00911898). Finally, a dose-ascending study of MM-111 in combination with multiple treatments studied 86 patients with advanced HER2-positive cancers and concluded a phase II dose of 20 mg/kg every week or 40 mg/kg every 3 weeks [52].

#### 3.4.3. Izalontamab (SI-B001)

SI-B001 is a bispecific antibody that targets the EGFR and HER3 signaling pathways and has shown encouraging efficacy in preclinical studies. Xue et al. predicted human pharmacokinetics and clinically effective doses of SI-B001 depending on the cancer type [59]. A phase I study is currently underway to determine the safety and tolerability of SI-B001 in patients with epithelial malignancies to further determine the recommended phase II doses for subsequent studies (NCT04603287). Meanwhile, A phase II study of 55 patients with locally advanced or metastatic EGFR/ALK wild-type NSCLC who had failed first-line anti-PD-1/L1 therapy, with or without platinum-based chemotherapy, found that SI-B001 plus docetaxel demonstrated antitumor activity. Of the 55 patients, 48 were evaluable for efficacy; the ORR was 31.3% (15/48, [95%CI] 18.7–46.3), and the DCR was 77.1% (37/48, [62.7–88.0]). Of the 45 pts enrolled in Cohort B, 38 were evaluable for efficacy. Among 22 evaluable pts in schedule 1, the ORR was 45.5% (10/22, [24.5, 67.8]), and the DCR was 68.2% (15/22, [45.1, 86.1]). The most common grade ≥ 3 treatment-related adverse events (TRAEs) were myelosuppression (17%), decreased neutrophil count (15%), and decreased white blood cell count (12%). There was no drug-related death [60].

#### 3.4.4. Zanidatamab (ZW25)

Zanidatamab is a bispecific antibody developed by Zymeworks using the Azymetric platform to bind two non-overlapping epitopes of the HER2 receptor. A phase I trial evaluated zanidatamab in patients with pre-treated advanced HER2-expressing cancers with and without chemotherapy (NCT02892123). Results from cohort 1 patients with breast cancer who received zanidatamab and docetaxel were published. Among 22 efficacy evaluable patients, zanidatamab showed an ORR of 86.4%, a 6-month PFS of 90.9%, and a manageable safety profile [61]. The final results from this trial, which also studied the efficacy of zanidatamab in the HER2-expressing biliary tract, as well as in colorectal and gastro-esophageal cancers, confirmed an ORR of 37% and a favorable safety profile in patients treated with zanidatamab alone [62].

Recent data from a phase II study showed that zanidatamab with standard chemotherapy was a highly active treatment regimen for first-line therapy of HER2-positive metastatic gastro-esophageal adenocarcinoma (NCT03929666). Among 42 patients, a tolerable safety profile with an 18-month survival rate of 87.3%, ORR of 79%, DCR of 92%, and a median DoR of 20.4 months were reported [63]. Finally, HERIZON-GEA-01, a phase III trial that is currently underway, aims to evaluate zanidatamab with tislelizumab (anti-PD-1 monoclonal antibody) for HER2+ gastro-esophageal adenocarcinoma (NCT05152147) [58].

### 3.5. CEA Targeted

#### MEDI-565 (MT111, AMG211)

MEDI-565 is a BiTE-bispecific antibody that targets the CD3 and human carcinoembryonic antigen (CEA) [64]. CEA has been demonstrated to be an important prognostic factor for colorectal cancer and is also overexpressed in other solid tumors [65,66]. The concentration of circulating CEA has also been shown to be a supportive diagnostic marker in both NSCLC and SCLC, and the receptor carcinoembryonic antigen-associated cell adhesion molecule 5 (CEACAM5) is considered a potential target [67]. A preclinical study demonstrated the efficacy of MEDI-565 in activating the T cells to eliminate CEA-positive tumor cells in vivo and in vitro [68]. A phase I dose escalation study of MEDI-565 in patients with advanced gastrointestinal malignancies showed an MTD of 5 mg, but unfortunately, no objective responses were observed in this study [69].

### 3.6. Immune-Checkpoint Targeted

#### 3.6.1. LY3434172

LY3434172 is an IgG-subtype bispecific antibody that targets the interaction between PD-1 (Programmed cell death protein 1) and PD-L1 (Programmed cell death ligand 1). The prevalence of PD-L1 expression is observed in approximately 24% to 60% of patients with NSCLC [70]. Preclinical studies have demonstrated bsAbs that simultaneously target both PD-1 and PD-L1 led to significant disinhibition of T-cells, hence driving their further development [71]. One arm of LY3434172 blocks the interaction of PD-1 to PD-L1/2, while the other arm blocks the binding of the ligand to CD80. The use of the drug led to considerable antitumor activity in mouse models, and a phase I study to evaluate its efficacy in solid metastatic cancers has been completed, and results are awaited (NCT03936959) [71].

#### 3.6.2. LY3415244

LY3415244 is a bispecific antibody that targets PD-L1 and T cell immunoglobulin and mucin domain-containing protein 3 (TIM3), another promising immunotherapeutic target. In a phase I clinical trial, LY3415244 was administered in patients with solid tumors, including lung cancer. Due to unexpected immunogenicity, the study had to be terminated, and antidrug antibodies were observed among the participants [72]. Other bispecific agents with similar PD-1/L1 and TIM-3 targeting include LB1410 (NCT05357651) and AZD7789 (NCT04931654), trials for which are currently ongoing.

#### 3.6.3. Erfonrilimab (KN046)

Erfonrilimab is an IgG subtype bispecific antibody that targets PD-L1 and CTLA-4 and has been tested in multiple trials with different types of malignancies. In a phase I clinical trial, erfonirilimab was administered to patients with NSCLC and nasopharyngeal cancer who had progressed on immune checkpoint-based immunotherapy. An ORR of 12% and a DCR of 52% were observed, with a tolerable adverse effect profile (NCT03529526) [73]. Preliminary results from a phase II trial of erfonirilimab in combination with nab-paclitaxel for treating triple-negative breast cancer showed a decent safety profile, with a 12-month progression-free survival (PFS) and OS of 38.3% and 80%, respectively [74].

#### 3.6.4. Bintrafusp Alfa (M7824)

Bintrafusp alfa is a bifunctional fusion protein comprising the extracellular domain of the human TGF-β receptor fused with the heavy chain of an IgG antibody blocking PD-L1, thereby inhibiting both pathways [75,76]. In preclinical models, it has led to significant tumor regression and reverse epithelial–mesenchymal transition in human cancer cell lines [77]. In a phase I study, among patients with NSCLC refractory to immune checkpoint inhibitors, bintrafusp alfa showed a manageable safety profile and an ORR of 4.8% (NCT02517398) [78]. An expansion cohort of the trial included 80 patients, and an ORR of 21.3% was observed [79].

#### 3.6.5. SHR-1701

SHR-1701 is a novel bifunctional fusion protein consisting of a monoclonal antibody against PD-L1 fused with the extracellular domain of TGF-β. It has been utilized in a phase I dose-escalation study in patients with advanced NSCLC with EGFR mutation after treatment failure with at least one EGFR TKI. A manageable safety profile was observed, a recommended phase two dose of 30 mg/kg every three weeks was determined, and an ORR and DCR were noted to be 16.7% and 50%, respectively (NCT03774979) [80]. A phase II study evaluated neoadjuvant SHR-1701 with or without chemotherapy for three cycles, followed by surgery or radiotherapy in stage III NSCLC patients. SHR-1701 ± chemotherapy demonstrated encouraging anti-tumor activity with the best ORR of 70.1%, an event-free survival (EFS) of 18.2 months, and a tolerable safety profile [81].

#### 3.6.6. Tebotelimab (MGD013)

Tebotelimab is an investigational bispecific antibody that belongs to the DART family. It is designed to block PD-1 and LAG-3 checkpoint molecules to restore exhausted T cells for killing tumor cells. It was combined with margetuximab, an anti-HER2 monoclonal antibody, in a phase I study among patients with advanced HER2+ malignancies (NCT03219268). The drug demonstrated an acceptable safety profile and good anti-tumor activity with an ORR of 40% [82]. Phase I/II study showed encouraging results among patients with refractory hepatocellular carcinoma (NCT04212221). In patients with prior exposure to immune checkpoint inhibitors, tebotelimab showed an ORR of 3.3%, a DCR of 50%, and a median PFS of 2.4 months. Immune checkpoint inhibitor naïve patients had 13.3% ORR and a 46.7% DCR [83].

#### 3.6.7. MEDI5752

MEDI5752 is a bispecific human IgG1 monoclonal antibody developed by AstraZeneca and targets the PD-1 and CTLA-4 receptors. It has been evaluated in combination with platinum chemotherapy among patients with NSCLC in a phase Ib/II clinical trial, and the data were presented at ESMO 2022. MEDI5752 led to an ORR of 55.6% and a median PFS of 13.4 months in the PD-L1 < 1% population with an acceptable safety profile [84].

#### 3.6.8. AZD2936

AZD2936 is a bispecific, humanized IgG1 targeting PD-1 and TIGIT. The first human trial, ARTEMIDE-01 (NCT04995523), evaluated the drug in open-label multicenter research, including advanced NSCLC patients with a PD-L1 TPS ≥ 1%. It examined dosage escalation, expansion safety, tolerance, dose-limiting toxicities, and preliminary effectiveness [85].

In total, 80 pts were enrolled; 72.5% of patients had adenocarcinoma, 96.3% had metastases, and 22.5% had brain metastases. The therapy median duration was 11 weeks. AZD2936 was well-tolerated [85]. Pruritus, rash, and lipase rise were the most prevalent treatment-related adverse events (TRAEs) (6.3% each). Immune system problems, acute hepatitis, and fatigue (*n* = 1) were serious TRAEs in three patients (3.8%). A total of 3 partial responses and 30 stable diseases were seen in 76 evaluable patients [85]. A randomized dosage optimization cohort is investigating AZD2936 in CPI-naïve NSCLC patients.

### 3.7. VEGF Targeted

#### 3.7.1. Ivonescimab (AK112)

Ivonescimab is a bispecific antibody that targets PD-L1 and vascular endothelial growth factor (VEGF) and prevents the interaction of both ligands with their respective receptors. VEGF expression has been reported in more than 70% of lung cancers, with even higher expression in the adenocarcinoma subtype. Ivonescimab received a breakthrough therapy designation (BTD) from The Center for Drug Evaluation of the China National Medical Products Administration in combination with docetaxel for the treatment of patients with locally advanced or metastatic NSCLC who had been unresponsive to prior PD-L1 inhibitor plus platinum-based doublet chemotherapy. In addition, the drug has also received a BTD as a first-line treatment for patients with locally advanced or metastatic PD-L1-positive NSCLC.

Results from the phase 1b/II trial showed ivonescimab monotherapy to be safe in advanced NSCLC, with an adverse event rate of 13.5%. Regarding efficacy, when doses of >10 mg/kg were administered every three weeks, the ORR and DCR were 42.9% and 92.9%, respectively, in 56 evaluable patients [86]. Another phase II trial comprised three cohorts, and the efficacy of ivonescimab was investigated in combination with chemotherapy in advanced NSCLC. In the cohort with previously untreated patients with NSCLC without EGFR/ALK alterations (cohort 1), patients with squamous histology experienced a 75% ORR with a median DOR of 15.4 months and 95% DCR, and the 9-month PFS and OS rates were 67% and 93%, respectively. Meanwhile, those with non-squamous histology experienced a 55% ORR, DOR was not reached, DCR was 100%, and the 9-month PFS rate and OS rates were 61% and 81%, respectively [87,88]. A phase III trial of ivonescimab or placebo in combination with chemotherapy is ongoing in EGFR-mutated, advanced NSCLC patients who failed a prior EGFR TKI (NCT05184712).

#### 3.7.2. PM8002

PM8002 is a bispecific PD-L1/VEGF-A antibody. Phase 1b and phase IIa dosage escalation and dose expansion studies used 20 mg/kg q2 or 30 mg/kg q3 as monotherapy (recommended phase II levels) (ChiCTR2000040552 clinical study) [89]. As of February 2023, 263 solid tumor patients with 0–4 systemic therapies were included in the trial, reported at ASCO 2023. A total of 27 NSCLC patients with EGFR mutations had an ORR of 18.5% (5/27) [89]. Proteinuria, hepatitis, and hypertriglyceridemia caused 14% of patients to quit therapy. In advanced solid tumor patients, it exhibited promising anticancer efficacy and safety [89]. PM8002 monotherapy and chemotherapy combination studies are underway for several indications.

#### 3.7.3. IMM2510

IMM2510 is a PD-L1 and VEGF inhibitor. Individuals with advanced solid tumors who had previously undergone systemic therapy were excluded from a phase-1 open-labeled trial if they had received two prior lines of PD-1/PD-L1 or antiangiogenic inhibitor medicines (chiCTR20211203 clinical study). Preliminary data were presented at ASCO 2023; by December 2022, 22 patients had received IMM2510 at various dosages [90]. A pilot trial with 17 evaluable patients showed safety and antitumor efficacy, including one confirmed PR in lung squamous cell carcinoma after 24 weeks [90]. A phase I/II study is currently ongoing.

#### 3.7.4. HB0025

HB0025 is a humanized recombinant anti-PD-L1 monoclonal antibody VEGFR1 fusion protein. It was evaluated in a dose-escalation experiment in patients with advanced solid tumors who had previously received several lines of systemic therapy (NCT04678908) [91]. The trial was presented at ASCO 2023, and 30 patients were enrolled as of August 2022, with 22 evaluable [91]. Preliminary results revealed a 9% (2/22) ORR and disease control in 50% of patients. A total of 83% of patients had treatment-related side effects. It had a tolerable safety profile, with no treatment-related fatalities [91]. It also showed anti-tumor effectiveness and a larger expansion cohort is now being examined at 10 mg/kg dosages.

#### 3.7.5. Dilpacimab (ABT-165)

Dilpacimab is a DVD-Ig subtype of bispecific antibody that targets both the VEGF and DLL4 pathways. It was developed by AbbVie and was used in a phase I dose-escalation study among patients with advanced solid tumors (NCT01946074). The preliminary efficacy data reported partial responses in 10.9% of the patients, with the remaining patients having either stable (52.7%) or progressive disease (12.7%) [92].

## 4. Challenges and Future Prospects

The research and development in the field of bispecific antibodies and their utilization as drugs to treat malignancies have been ongoing at a rapid pace. However, certain limitations must be addressed before bispecific antibodies’ clinical application can be recommended. The Fc-fragment of IgG-type antibodies is immunogenic and has the potential to bind to Fc-receptors on other effector cells, thereby leading to tumor cell-independent T-cell activation, which is responsible for cytokine release syndrome and other complications. Furthermore, the logistical challenges of bispecific administration must also be considered.

Finally, due to their synthetic nature, they are immunogenic and possess the potential to elicit an immune response in the body, which emphasizes the importance of designing them with a refined humanized component. Bispecific antibodies are difficult and expensive to manufacture. This can limit their availability and make them unaffordable for some patients. The development of new technologies to synthesize bispecific antibodies with a range of affinity in preclinical and clinical trials will improve their safety and clinical efficacy in the future. With these newer drugs, there are some unique adverse effects to be expected. For instance, patients receiving amivantamab have a high chance of infusion type reactions. Most of the other therapies for lung cancer are in the early stages of the trials, and drug-specific adverse effects will be reported in the future when widely used in clinical settings.

## 5. Conclusions

With the identification of novel potential targets and the development of biotechnology and manufacturing practices, the application of bispecific antibodies in treating solid tumors has shown significant progress. Over 200 antibodies have shown noteworthy preclinical results and have been tested in various clinical trials. There is a need for more extensive phase II/III trials to validate the results of bispecific antibodies observed in preclinical and early-phase clinical trials, which will lead to the widespread application of these drugs to treat solid tumors. The challenges that need to be addressed with the use of bispecific antibody-based therapy include off-target binding, manufacturing difficulties, low efficacy, immunogenicity, cost, and resistance mechanisms. It is imperative to carefully select the precise format and a rational combination of target antigens to achieve the optimal therapeutic effect of bispecific antibodies.

## Figures and Tables

**Figure 1 pharmaceuticals-16-01461-f001:**
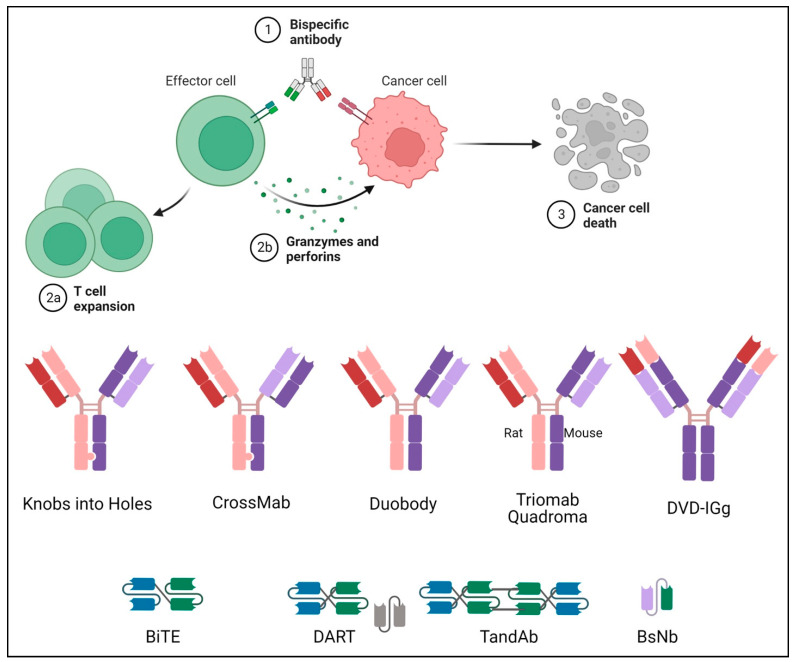
Mechanism and structure of different types of BsAbs. The basic mechanism of a bispecific antibody is explained in the upper portion with the steps in activation of the T cell to cancer cell death number from 1 to 3. The lower portion is a pictorial representation of the basic structures of different bispecific antibodies.

**Figure 2 pharmaceuticals-16-01461-f002:**
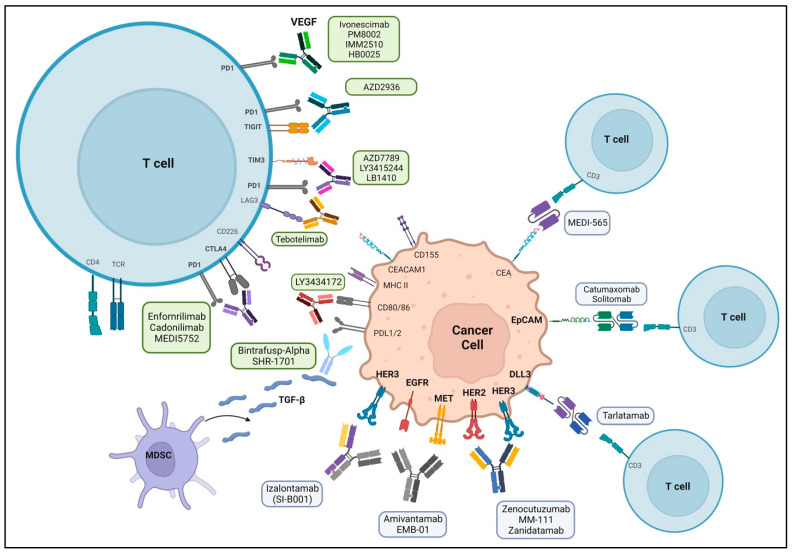
Bispecific antibodies and their target sites in lung cancer therapeutics. Bispecific antibody targets include sites in the tumor cell or an interaction with tumor cell and T-cell. Amivantamab is the only FDA-approved therapy, and other bispecific antibodies are in various stages of clinical trials.

**Table 1 pharmaceuticals-16-01461-t001:** Approved Bispecific Antibodies in Oncology.

S No.	NCT	N	Phase	Target	Therapy	Trade Name	Indication	Approval Date	Approved by
1	NCT00836654	24	I/II	CD3/EpCAM	Catumaxomab	Removab	Malignant Ascites	April 2009	EMA
2	NCT01209286	36	II	CD3/CD19	Blinatumomab	Blincyto	Relapsed/Refractory B-cell ALL	December 2014	FDA
3	NCT02609776	362	I	MET/EGFR	Amivantamab	Rybrevant	EGFR Exon20ins NSCLC	May 2021	FDA
4	NCT03070392	378	II	CD3/TCR	Tebentafusp-tebn	Kimmtrak	Metastatic uveal melanoma	January 2022	FDA
5	NCT02500407	238	I/II	CD20/CD3	Mosunetuzumab	Lunsumio	Relapsed/Refractory B-cell lymphoma	June 2022	FDA
6	NCT04868708	45	I	PD-1/CTLA-4	Candonilimab		Recurrent/Metastatic cervical cancer	June 2022	NMPA
7	NCT03145181 NCT04557098	165	I/II	CD3/BCMA	Teclistamab-cqyv	Tecvayli	Relapsed/Refractory Multiple Myeloma	October 2022	FDA
8	NCT03625037	148	I/II	CD3/CD20	Epcoritamab-bysp	Epkinly	Relapsed/Refractory DLBCL	May 2023	FDA
9	NCT03075696	132	I/II	CD3/CD20	Glofitamab-gxbm	Columvi	Relapsed/Refractory DLBCL or large B-cell lymphoma	June 2023	FDA
10	NCT04649359	97	II	CD3/BCMA	Elranatamab-bcmm	Elrexfio	Relapsed/Refractory Multiple Myeloma	August 2023	FDA

Abbreviations: N—Number; CD—cluster of differentiation; cMet—C-mesenchymal-epithelial transition factor; CTLA-4—Cytotoxic T-Lymphocyte Antigen-4; EGFR—Epidermal Growth Factor Receptor; PD-1—programmed cell death 1; BCMA—B-cell maturation antigen; TCR—T-cell Receptor; EMA—European Medicines Agency; FDA—U.S. Food and Drug Administration; NMPA—National Medical Products Administration, DLBCL—diffuse large B-cell lymphoma; BCMA—B-cell maturation antigen.

**Table 2 pharmaceuticals-16-01461-t002:** Clinical Trials Utilizing Bispecific Antibodies in Lung Cancer.

S No.	NCT	Study Name	Estimated Number	Phase	Target	Therapy [Platform]	Start Date	Completion Date
1	NCT02609776	CHRYSALIS	780	I	MET/EGFR	Amivantamab [Duobody]	May 2016	January 2024
2	NCT04606381	PALOMA	196	Ib	MET/EGFR	Amivantamab	November 2020	October 2024
3	NCT04077463	Chrysalis-2	460	I/Ib	MET/EGFR	Amivantamab + Lazertinib	September 2019	March 2026
4	NCT04538664	PAPILLON	308	III	MET/EGFR	Amivantamab + Carboplatin + Pemetrexed	October 2020	January 2025
5	NCT04487080	MARIPOSA	1074	III	MET/EGFR	Amivantamab + Lazertinib	September 2020	November 2025
6	NCT04521179	-	30	II	HER2	KN026 [Charge Repulsion Induced Bispecific]	December 2020	October 2023
7	NCT02912949	eNRGy	250	I/II	HER2/HER3	Zenocutuzumab (MCLA-128) [Biclonics]	January 2015	December 2024
8	NCT03821233	-	174	I	HER2	ZW49 [Azymetric]	April 2019	August 2025
9	NCT02892123	-	279	I	HER2	ZW25 (Zanidatamab) [Azymetric]	September 2016	August 2023
10	NCT03261011	-	153	1a/1b	PD-1/CTLA-4	Cadonilimab (AK104) [Tetrabody]	October 2017	September 2020
11	NCT04647344	-	60	Ib/II	PD-1/CTLA-4	Cadonilimab (AK104)	November 2020	April 2023
12	NCT04646330	-	114	Ib/II	PD-1/CTLA-4	Cadonilimab (AK104) + Anlotinib	November 2020	December 2023
13	NCT04544644	-	30	II	PD-1/CTLA-4	Cadonilimab (AK104) + Anlotinib	September 2020	September 2023
14	NCT03819465	MAGELLAN	258	Ib	PD-1/CTLA-4	MEDI5752 [Duetmab]	December 2018	March 2026
15	NCT03838848	-	120	II	PD-1/CTLA-4	KN046 [single-domain antibody]	May 2019	Terminated
16	NCT04054531	-	50	II	PD-1/CTLA-4	KN046 + Platinum chemotherapy	September 2019	June 2021
17	NCT04474119	ENREACH-L-01	482	III	PD-1/CTLA-4	KN046 + Paclitaxel + Carboplatin	September 2020	August 2023
18	NCT04900363	-	108	Ib/II	PD-1/VEGF	AK112 [Tetrabody]	May 2021	May 2024
19	NCT04995523	ARTEMIDE-01	192	I/II	PD-1/TIGIT	AZD2936	September 2021	July 2025
20	NCT04931654	-	81	I/IIa	PD-1/TIM-3	AZD7789	September 2021	July 2025
21	NCT02324257	-	149	I	CD3/CEA	RO6958688 [CrossMab]	December 2014	September 2019
22	NCT02650713	-	228	Ib	CD3/CEA	RO6958688 + Atezolizumab	January 2016	January 2020
23	NCT03337698	Morpheus Lung	435	Ib/II	CD3/CEA	RO6958688	January 2018	August 2025
24	NCT01221675	-	18	I/II	HSG/CEA	TF2 (IMP288)	June 2011	April 2016
25	NCT04822298	-	3	I	CD3/PSMA	Acapatamab (AMG160) [BiTE]	August 2021	January 2022
26	NCT04496674	-	86	I	CD3/PSMA	CC-1	February 2022	September 2025
27	NCT04750239	-	3	I/II	CD3/GD2	Nivatrotamab	August2021	April 2022

Abbreviations: N—Number; CD3—cluster of differentiation 3; CEA—Carcino-Embryonic Antigen; cMet—C-mesenchymal-epithelial transition factor; CTLA-4—Cytotoxic T-Lymphocyte Antigen-4; EGFR—Epidermal Growth Factor Receptor; HER2—Human Epidermal Growth Factor Receptor 2; HER3—Human Epidermal Growth Factor Receptor 3; PD-L1—programmed death-ligand 1; PD-1—programmed cell death 1; HSG—human serum albumin; PSMA—prostate-specific membrane antigen; TIGIT—T cell immunoreceptor with Ig and ITIM domains; TIM-3—T cell immunoglobulin and mucin-domain containing-3; GD2—Disialoganglioside.

## Data Availability

Data sharing is not applicable.

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
