# Peer review of "Bispecific Antibodies in Lung Cancer: A State-of-the-Art Review"

_pharmaceuticals, 2023, doi:10.3390/ph16101461_

Round 1

Reviewer 1 Report

The presented review article summarizes the mechanism-of-action and structure subtypes of bispecific antibodies. Also, it goes over the current bispecific antibody candidates in the treatment of lung cancers and their corresponding targets. Overall, this is a comprehensive review manuscript that would attract some interest in the field of lung cancer therapy utilizing bispecific antibodies. However, reorganization is needed. I would suggest acceptance after major revision with the following comments:

Major points:

1. The authors need to re-number the sections more clearly, like 1. Introduction; 2. Mechanism of action; 3. Structure of bispecific antibodies; 4. Application in lung cancer; and then number the sub-section as 1.1, etc. The “mechanism of action” section could be combined with the “application in lung cancer” as targets/MOA is also categorized in this section. Also, it would be better to shorten the “structure of bispecific antibodies” section to further emphasize the focus of the manuscript. It’s more important to explain the advantages/disadvantages of each subtype and may include the subtype information in Table 2 to demonstrate the connection between content.

2. In the “Application in lung cancer” section, a short summary before introducing different targets will help readers follow the logical flow. It would be better to put Figure 2 as a summary figure as it includes many different MOAs of bispecific antibodies besides amivantamab. A detailed legend would also help to understand this figure with numbering the different MOAs. Table 2 is also a summary table which should be mentioned in the summary paragraph instead of in the “amivantamab” section.

3. All the targets mentioned need to be shortly introduced (function, expression level in lung cancer and expression specificity).

4. The section “challenges and future prospects” is a bit too general. More detailed discussion regarding current limitations for different targets/candidates would be attractive to the readers as the authors provide a comprehensive summary of promising candidates in lung cancer treatment. What are the potential alternatives to produce bispecific antibodies?

Minor points:

5. The font/size of text needs to be consistent. Some of the antibodies are with the first letter in capital and some are not.

6. In the section “Amivantamab”, the introduction of “Lazertinib (3rd generation EGFR TKI)” should be mentioned for the first time when Lazertinib appeared in the manuscript. 

Reviewer 2 Report

The review article titled "Bispecific Antibodies in Lung Cancer: A State-of-the-Art Review" summarizes the current state of research focused on the development of bispecific antibodies for treating lung tumors. The topic is highly interesting as it is very current and still has a long way to go. However, I have some doubts about the specific contribution of this review study. Here are some of my major concerns:

  1. While the topic is novel, there are already some existing reviews in the literature, such as the one published recently (DOI: 10.3390/ijms24129855). What does this new review provide that hasn't been mentioned before? This aspect should be clarified or improved upon to differentiate it from the previously published review in one of the MDPI journals.
  2. The sections on mechanisms of action and bispecific antibody structure are poorly described. The effective use of Figure 1 is lacking.
  3. Figure 1 needs further refinement and clearer figure captions. As it is presented, it is confusing and lacks sufficient information.
  4. In the following sections where the different types of antibodies generated are described, it is not clear which have already been tested in lung cancer and which in other types of tumors. This should be clarified and those that could have an impact on lung treatment should also be recommended if they have not yet been tested for this type of tumor.

Round 2

Reviewer 1 Report

The number in section 2 should be 2.x.x instead of 1.x.x.

Reviewer 2 Report

I believe the modifications made in the new revision, as well as the responses provided to my comments, are sufficient to warrant the publication of this review. I have no further comments.